# Complete Real-Scale Application of Recycled Aggregates in a Port Loading Platform in Huelva, Spain

**DOI:** 10.3390/ma13112651

**Published:** 2020-06-10

**Authors:** Francisco Agrela, Francisco González-Gallardo, Julia Rosales, Javier Tavira, Jesús Ayuso, Manuel Cabrera

**Affiliations:** 1Leonardo Da Vinci, Campus Rabanales, University of Córdoba, 14014 Córdoba, Spain; 2Area of Construction Engineering, University of Cordoba, 14014 Cordoba, Spain; ir1gogaf@uco.es (F.G.-G.); jrosales@uco.es (J.R.); jtavira@ciccp.es (J.T.); ir1ayuje@uco.es (J.A.); manuel.cabrera@uco.es (M.C.)

**Keywords:** seaport loading platform, recycled aggregates, civil infrastructures, structural granular layers, construction and demolition waste

## Abstract

The application of recycled aggregates (RA) from construction and demolition waste and crushed concrete blocks is a very important challenge for the coming years from the environmental point of view, in order to reduce the exploitation of natural resources. In Spain, the use of these recycled materials in the construction of road bases and sub-bases is growing significantly. However, presently, there are few studies focused on the properties and behavior of RA in civil works such as road sections or seaport platforms. In this work, two types of RA were studied and used in a complete real-scale application. Firstly, recycled concrete aggregates (RCA) were applied in the granular base layer under bituminous superficial layers, and secondly mixed recycled aggregates (MRA) which contain a mix of ceramic, asphalt, and concrete particles were applied in the granular subbase layer, under the base layer made with RCA. Both RA were applied in a port loading platform in Huelva, applying a 100% recycling rate. This civil engineering work complied with the technical requirements of the current Spanish legislation required for the use of conventional aggregates. The environmental benefits of this work have been very relevant, and it should encourage the application of MRA and RCA in civil engineering works such as port platforms in a much more extended way. This is the first and documented real-scale application of RA to completely build the base and sub-base of a platform in the Huelva Port, Spain, replacing 100% of natural aggregates with recycled ones.

## 1. Introduction

Several studies have demonstrated the feasibility of using recycled aggregates (RA) from construction and demolition waste (CDW) in structural road layers over last two decades [1,2]. Therefore, in recent years, the use of recycled aggregates in road and concrete applications has advanced considerably [3,4,5,6,7,8]. Although research focused on RA started in the 1990s, there are not enough specific technical regulations to promote the use of these types of aggregates [9]. This is why in Spain the Structural Concrete Instructions EHE-08 [10] are used for concrete construction, and the General Technical Specifications for Construction of Roads and Bridges [11] are used for road construction. Based on the proportions of the components of RA, De Brito, Agrela, and Silva (2019) [12], proposed a new classification of these that can be divided into:-Recycled concrete aggregates (RCA): recycled aggregates which contain concrete particles larger than 85% or 90% of total dry mass. It would be possible to divide this category in two types, I and II (Table 1), because there are some samples of RCA with more water absorption capacity and other different values in their properties.-Mixed recycled aggregates (MRA): in this category there are three types, and only types I and II can be used in road layers (Table 1). These MRA-I and MRA-II are products obtained in the treatment of CDW, containing ceramic particles (Rb) between 15% and 40%.

Other types are included in this proposal, MRA-III and RAA, but Table 1 only includes the most feasible RA to be used in road pavements, bases, sub-bases, and capping of esplanades, and additionally, showing the different applications of RA on roads.

An important value is that obtained with the Los Angeles Abrasion Test, which consists of producing an abrasive action by using standard steel balls that, when mixed with aggregates and rotate in a drum during a specific number of revolutions, also cause impact on aggregates. The percentage of wear of the aggregates due to friction with steel balls is determined and is known as the Los Angeles abrasion value, it is the difference in size before and after abrasion. The lower the percentage, the better the abrasion performance of the material.

RCA are the most widely used in granular unbound road layers. This material meets the requirements establish by the PG-3, usually showing the same characteristics as a natural aggregate [13].

Vegas et al. (2011) [14] conducted a pre-normative study to determine the physical, chemical, and mineralogical properties of MRA for their use in structural and unbound road layers. This study concluded that MRA with variable contents of concrete and ceramic particles could produce pozzolanic reactions, increasing the long-term mechanical properties of the structural road layer.

In recent years, RA have been applied in different studies for use on low-traffic roads and paths. MRA are generally used in road layers with reduced structural requirements, such as rural or pedestrian paths [13] and RCA are used in more demanding applications, such as on roads with a higher traffic volume [2]. Commonly, the different types of RA are partially applied. However, it is necessary to use these recycled materials in complete applications, in which all the conventional aggregates applied in structural layers can be replaced by RA to produce a significant reduction in environmental impact, maintaining the appropriate mechanical behavior and bearing capacity with respect to the conventional layers.

In this work, RCA and MRA have been applied in structural layers of seaport platforms. The RA have been applied over the entire surface of two layers the structural section. The design of port pavements is carried out according to their use and future activity, establishing the different calculation loads, their intensity of use and the type of existing subgrade. Specifically, in Spain, the recommendations in ROM 4.1-94 [15] and Standard 6.1-IC [16] are used for the design of port pavements.

In this work, two layers of 0.25 m of natural aggregates form of quarries were replaced by recycled aggregates for the port sub-base platform construction. Mechanical behavior, leaching properties, equivalent module, and deflections were studied to demonstrate the feasibility of using these materials in this type of application, which reduces the consumption of natural resources or raw materials, decreasing the ecological footprint.

## 2. Research Design

The current research aims to study and evaluate the physical, chemical, and mechanical behavior of recycled concrete aggregates (RA) for their use in a complete real-scale application in structural layers on a loading platform at the port “Ciudad de Palos” in Huelva. This platform is located in the area known as the outer harbor, on the left bank of the Ria Huelva (Huelva, Spain) and has an unpaved surface area of 28,500 m^2^. An area of 8200 m^2^ was chosen to develop this research (Figure 1).

This study was divided into several stages, which are described below (Figure 2) in the chronological order that was followed during the research.
-Task 1. Study of MRA and RCA properties. In this initial phase, recycled aggregates (RCA and MRA) produced at the treatment plant were evaluated, studying the process of treatment applied at the recycling plant, the fines particles content, physic-chemical properties, or the potential contamination by leaching of the recycled aggregates. At the end of this phase, a solution was proposed to be applied in the real-scale application.-Task 2. Mechanical behaviour of MRA and RCA was studied in order to establish the possibilities of using these recycled aggregates in the real-scale application. In this part of the research, compaction and bearing capacity were determined.-Task 3. Execution of the loading platform at the Port of Huelva, specified in Figure 1, applying MRA and RCA in the base and sub-base structural layers of the platform.-Task 4. Quality control of the execution processes, verifying that the specifications previously established and corroborated by laboratory tests were met. In this phase, density and water absorption in both granular structural layers of the loading platform were determined to confirm the correct application of MRA and RCA.-Task 5. Medium- and long-term auscultation tests, using the application of an impact deflectometer. This study was carried out in the entire area of the loading platform, which was divided into 14 streets with a width of 5 m, numbered and delimited by measuring equipment. This allowed the study of the area of a longitudinal direction, obtaining an equivalent stretch of 1750 m in length.-Task 6. Final evaluation of the application. After the execution of the application, and with a defined frequency, the basic parameters were evaluated.

This study has been developed in accordance with the technical specifications regulated by current legislation, PG3 and ROM-4.1/94, for both quality control of the materials, design and appropriate execution of the loading platform.

## 3. Recycled Materials

In this work, two types of recycled aggregates (RCA and MRA) were studied to be used as sub-base and base layers in the port loading platform.

Due to the appropriate mechanical properties of the recycled aggregates for their use in road layers compared to artificial gravel (artificial gravel (AG) is an aggregate of granular particles formed by stone material, it originates from the fragmentation of artificially crushed rocks), the authority of the Port of Huelva decided to modify a part of the structural section of the loading platform called “Ciudad de Palos”. Two layers of AG were replaced by MRA and RCA. To carry out this replacement, characterization studies were initially performed on both RA, and the mechanical behavior was analyzed, comparing their results with AG (conventional materials in road construction).

On the one hand, mixed recycled aggregates (MRA) obtained from the treatment of construction and demolition waste (CDW) with variable contents of ceramic particles (Rb), in this case less than 30% were studied. The MRA used was composed of other different elements, including asphalt, particles with adhered mortar, natural aggregates, glass, gypsum, and other impurities. This material was used to make the sub-base in the loading platform section, and was classified as MRA-II, because the content of gypsum and other elements were reduced to less than 5%. It is very important to make an adequate selection at source of the CDW to achieve very low contents of different particles, concrete (Rc+Ru), ceramic (Rb), or even asphalt (Ra).

On the other hand, recycled concrete aggregates (RCA) were used and were composed of more than 85% particles of concrete and natural aggregates. This material was used to build the base of the loading platform. In this case it was classified as RCA-II, due to the content of natural aggregates and particles with adhered mortar, which are greater than 85% but less than 90% required to achieve the RCA-I type. It was possible to observe that gypsum particles and others such as plastic, glass, etc., presented less than 1%, resulting a suitable material with few impurities.

Both materials were obtained from different treatment processes at a recycling plant located in Huelva, Spain according to EN: 13242: 2003 (Aggregates for unbound and hydraulically bound materials for use in civil engineering).

Table 2 shows the results of the composition of recycled aggregates according to EN 933-11. In addition, they were classified in accordance with Table 1, proposed by de De Brito, Agrela, and Silva (2019) [12].

The application of RCA and MRA on structural layers of the roads or port loading platforms could mean an improvement in its mechanical properties in the medium and long term [17]. Other authors demonstrated the self-cementation capacity of RCA because the non-hydrated cement particles existing in the old concrete particles, when entering in contact with water in their second cycle of life, could activate different chemical reactions of hydration, contributing to an improvement in the bearing capacity of the esplanade [18,19].

Table 3 shows the information concerning the properties tested in the laboratory according to the standards indicated.

PG-3 requires a maximum content of water-soluble sulfates for recycled aggregates from concrete demolitions (limit value 0.7%), with the results that both MRA and RCA do not exceed this value. As for the results of the tests of sand equivalent and slab index of both samples (RCA and MRA), the requirements established by the Spanish standard PG-3 [11] for fill of graded crushed aggregate are fulfilled.

The values of water absorption are slightly higher in the MRA samples with respect to RCA, which has been taken into account in the process of compaction of the layers.

On the other hand, the results obtained from the Los Angeles coefficient in both samples are slightly higher than the limit value of 35 that establishes the regulation for natural arid, being greater than the limit for recycled aggregates (<40).

The particle size distribution curves of both materials, RCA and MRA, are represented in Figure 3 together with the corresponding particle size distribution envelope of the artificial axes ZA-0/20 (granular mixtures) as indicated in PG-3 [11]. It can be seen that these curves present a particle size distribution within limits.

Based on the results obtained, RCA present better chemical and physical properties than MRA as it is usual. In general, both RA have properties similar to AG named in the Spanish standard as ZA-0/20.

## 4. Laboratory Tests and Results

### 4.1. Proctor Test/Moisture–Density Relationship

Two different types of tests were performed to identify the relationship between density and humidity of the different levels. For the subgrade we considered the standard Proctor test (PN), while for the subbase and base we used the modified Proctor test (PM).

The subgrade was constructed with selected soil, being necessary to obtain in situ 100% or more of dry density standard Proctor determined in laboratory according to UNE 103500:1994. The data for optimum moisture and maximum dry density were 5.5% and 1.79 ton/m^3^ respectively.

The modified Proctor (MP) was performed according to UNE 103501:1994. This test is similar to the standard Proctor but in this case five layers of the soil were compacted in the modified Proctor mold, applying a higher compaction energy.

The data for maximum dry density and optimum moisture are shown in Figure 4.

The maximum density and optimum moisture content obtained for the layer built with RCA were 2.08 ton/m^3^ and 10.2% respectively. The MRA series presented a lower density than the RCA (2.04 ton/m^3^ and optimum humidity of 10.5%) samples mainly because the MRA showed a lower value of saturated dry surface density with respect to the RCA as shown in Table 2, according to Poon and Chan (2006) [1].

The high porosity found in MRA, especially in the mortar and ceramic materials, is the reason why the optimum moisture content of compaction is slightly higher than in the RCA series.

### 4.2. California Bearing Ratio (CBR)

The CBR method measures the shear strength of a soil or aggregate compacted in a laboratory under controlled conditions of density and moisture and is used to evaluate the bearing capacity of the soil as subgrade, subbase, and base of a port and a road pavements as well as land classification. The CBR value was determined according to EN 13286-47: 2012, and this test depends on moisture, density, and overload conditions.

Table 4 shows the results obtained in CBR tests for the three materials studied.

From the obtained values, the subgrade is classified as an E3 type (very good) with selected soils with CBR > 20.

The values obtained from the CBR index for both materials (RCA and MRA) are considered suitable for granular layers of pavements.

### 4.3. Triaxial Test

From this test method, resistant parameters of fine-grained soils and aggregates are obtained as well as the stress–strain ratio through the determination of confined compressive strength and shear stress.

The sample is first hydrostatically loaded until the confinement pressure is reached, then an axial load is applied which increases progressively until the sample breaks. The containment pressure is kept constant.

In this case, the triaxial tests have been carried out according to EN ISO 17892-9:2018, with four cycles of loading and unloading and a confining pressure of 55 KPa for all the cases, which have allowed for determination of the resistance and the deformation parameters of RCA and MRA.

For the MRA sample, the loading–unloading steps are carried out with maximum and minimum values of load deviation of 290 and 25 KPa respectively, corresponding to 75% and 6% of the maximum deviation of breakage (383 KPa) previously determined with other test samples.

In the same way, we proceed with the RCA samples, being the maximum and minimum values of load deviation of 483 and 30 KPa, corresponding to 67% and 4% of the maximum deviation of breakage (721 KPa). These results obtained for the MRA and RCA are shown in Figure 5.

Table 5 shows the secant deformation modules obtained for the MRA and RCA samples. It can be seen that the deformation modules remain practically constant from the second load cycle, there being a ratio of 5 and 4 respectively with the first load stage.

On the other hand, greater resistance and rigidity was observed in the RCA samples with respect to the MRA.

### 4.4. Leaching Test EN 12457-4:2004

An assessment procedure was carried out for heavy metals and inorganic anions present in RCA and MRA in accordance with the EU Landfill Directive. The test procedure consisted of a compliance test by leaching 90 g of dry sample in deionized water with a liquid/solid ratio of 10 (L/S = 10). The two materials (RCA and MRA) were classified according to the limits set by the Landfill Directive 2003/33/EC.

All samples were classified as inert, as can be seen in Table 6 compared to the legal limits.

RCA and MRA showed very low values of heavy metals and sulfates in the leaching test. The low sulfate content, which is not usual in recycled aggregates, is due to a selection at the source in the treatment plant that reduces very substantially the gypsum content in the recycled aggregate.

## 5. Real-Scale Application, Test Program, and Results

### 5.1. Description of the Sections

A surface semiflexible was designed, formed by granular layers and bituminous materials.

Granular layers are composed of a layer of 25 cm of mixed recycled aggregate (MRA) and 25 cm of recycled concrete aggregate (RCA).

The bituminous layer have a total thickness of 20 cm composed of a bituminous base 8 cm (AC basis G 22), an intermediate layer of 8 cm (AC bin 22 S) and a surface layer of 4 cm (AC 16 surf S) (Figure 6).

### 5.2. Compaction of Layers

Nuclear density equipment was used to determine the density and moisture content of the soil in situ, according to ASTM D-6938. This test method is a rapid non-destructive technique that is used as acceptance tests of compacted soil layers as long as the material under test is homogeneous.

Multiple measurements of density and moisture content were taken in surface of subgrade and granular layers (10 in subgrade, 40 in MRA, and 21 in RCA) to perform a statistical analysis of the results.

The mean and standard deviation (SD) values of each level are included in Table 7.

### 5.3. Deflection Measurements

Impact deflectometer testing is a method used to evaluate the support capacity of the subgrade, granular, and bituminous layers. The test involves the application of a dynamic load on a damper system that transmits to circular plate resting on the surface, allowing measurements of the deflections produced on its surface through several sensors aligned with the plate and adequate computer equipment according to ASTM D4694 [19].

The Dynatest HWD 8081 impact deflectometer was used for the development of the work according to General Technical Specifications for High-Performance Dynamic Monitoring Test [20]. This equipment records the vertical deflection of the surface under the point of application of the load and in six other points located at 30, 45, 60, 90, 120, and 150 cm respectively.

For the development of the work the study area has been divided into streets of 5 m wide, obtaining an equivalent length of 1750 m. Deflection measures have been taken every 10 m in all the layers that have been defined in Section 5.1 as long as it has been possible to measure them, taking into account the coordination of the equipment (deflectometers). At all test points, a previous impact of settlement has been made before the test impacts.

In Figure 7 a full aerial image of the experimental platform is presented, and Universal Transverse Mercator (UTM) coordinates at end and start of lanes are shown.

Deflection correction coefficients have been applied for the humidity of the subgrade and pavement temperature, according to the corrections established in Regulation 6.3-IC of the Ministry of Development and Standard NLT-338/07.

The tests were carried out on the finished surfaces (compacted and refined) of the subgrade, granular, and bituminous layers. Six months after completion of the work, in November 2014, a new measurement of the deflections on the tread layer was made to study its evolution.

Once the measurement work was completed, the results of the deflections in all its layers were achieved over a length of 710 m. Figure 8 and Figure 9 represent these measurements, and in Table 7 the mean values and standard deviation are included for each loading platform layer.

It can be seen that there is an improvement in the evolution of asphalt surface behavior: the mean deflection obtained in November is 23.5% less than compared to May, as well as an increase in uniformity as can be inferred from the standard deviation values.

According to the standard 6.3 IC: Rehabilitation of pavements, the completely finished pavement meets the requirements of homogeneous section and uniform behavior, obtaining an average value of deflections of 0.169 mm and a sample standard deviation of 0.045. On the other hand, the characteristic deflection obtained is 0.258 mm, which is equivalent to a 97.5% probability that the deflection will not be exceeded in the study section.

### 5.4. Equivalent and Inverse Modulus

Two different methodologies have been used to evaluating the structural capacities of the pavement from the deflections measured on the surface of the different layers: the equivalent modulus and modulus obtained by inverse calculation method.

On the one hand, the equivalent module Ev, which is obtained from the measurement of deflection that has been made of each layer and it represents an equivalent module value of all existing layers below the analyzed layer.

The bearing capacities of the various layers have been evaluated by the parameters of deflections and surface modules obtained with the first geophone according to the formulation proposed by Brown (1996).
(1)$$E_{v}=\frac{2pa\left(1-v^{2}\right)}{d}$$
where: *E_υ_*: Equivalent modulus of the entire pavement system beneath the load plate*a*: Radius of the FWD plate, 225 mm on granular layers and 150 mm on asphalt concrete.*pa*: Pressure of FWD impact load under the load plate, 246.KPa on granular layers and 693.21 KPa on asphalt concrete.*d*: Deflection at 0 mm of center of the FWD plate.*υ*: Poisson’s ratio.

The results of equivalent modulus in different layers were included in Table 8. The relative standard deviation (RSD) has been considered to compare the variability of the results between the different layers and dates (Figure 10, Figure 11 and Figure 12).

On the other hand, inverse calculation method is a backcalculation procedure commonly used to estimate the pavement layer moduli based on the non-destructive with impact deflectometer tests.

An elastic multilayer software called BISAR [21] was used. Theories of Burmister (1945) [22], and Acum and Fox (1951) [23], and a solution to determine stress and strain of Schiffman (1962) [24] are used in the algorithm of this software. Theoretical deflection was calculated for each layer and section using theoretical elastic modulus shown on Table 9. 

There is a significant evolution of the equivalent modulus on the completely finished asphalt surface: the mean of equivalent modulus obtained in November is 27.1% higher than those determined in May, as well as an increase in uniformity as can be inferred from the relative standard deviation.

Pavement instruction of Andalusia [11] shows a correlation between CBR and modulus, so subgrade moduli was determined from the values of the CBR tests of the existing soil. According to Huang (1993) [25] asphalt concrete have a 0.33 and granular layers and subgrade a 0.35 Poisson ratio.

Elastic moduli of the four layers shown were obtained using Evercalc [26]. The software recalculates the modules by an iterative process comparing the average data with the theoretical data. The process will run until it finds convergence with a limited error.

Table 9 shows the values of the modules obtained from the indirect calculation method and the results of some statistical parameters.

According to the results offered in Table 9, a decrease in the values obtained from the elastic modulus in the subgrade can be seen. A higher amount of rain registered in the area before the November test could be the reason for the decrease in the bearing capacity of this level. On the other hand, an increase of the elastic moduli of the MRA and RCA layers was observed (6.7% and 17.3% respectively), probably due to the cementation of some cement compounds existing in the materials of these layers. The elastic modulus of the asphalt layer has not changed significantly, although there is a slight increase in uniformity as can be seen in RSD values.

For all of the above, it can be concluded that the improvement of the equivalent modulus observed in the month of November was due to the increase in the bearing capacity of the MRA layer and especially the RCA layer.

Finally, it is necessary to emphasize the good bearing capacity that was obtained with the RCA and MRA layers. Effectively, RCA mean moduli (652 MPa) are greater than crushed stone moduli obtained from natural aggregates (500 MPa) and MRA mean moduli (380 MPa) are greater than natural selected soil (250 MPa) (Spanish general technical specifications for road construction, 2004).

## 6. Life Cycle Assessment

A study of the environmental impact caused by the production of each of the aggregates used in this study (MRA and RCA) was carried out by means of a life cycle assessment (LCA).

The aim was to check the environmental impact caused by the use of recycled aggregates instead of the methodology used in a traditional construction system. A comparative analysis of the production system used to produce MRA, RCA, and AG was carried out.

The environmental impact caused by the production process of each aggregate was quantified according to ISO 14040 (2006) and ISO 14044 (2006).

The LCA analysis was performed using the SimaPro^®^ 8.0.2 software application and processed using the CML-IA method [27]. The evaluation methodology EN 15804 + A1 (CEN 2013) was used for the study, as it is the most suitable methodology in relation to sustainability for building products and services.

The boundaries considered in each of the production systems of the three types of aggregates analyzed are shown in Figure 13. The infrastructures and production systems of the quarry and the RCA and MRA recycling plants have been considered. The equipment and machinery necessary for production are known.

The life cycle impact assessment methodology combines 11 impact categories. These impact categories are: abiotic depletion of elements, abiotic depletion of fossil fuels, global warming, ozone layer depletion, human toxicity, fresh water aquatic ecotox, marine aquatic ecotoxicity, terrestrial ecotoxicity, photochemical oxidation, acidification, and eutrophication.

According to the different aggregates production processes shown in Figure 13, an inventory phase was carried out in which inputs (energy and raw material) and outputs (products, co-products, wastes, and emissions) of the particular production system of each aggregate were considered. The Ecoinvent database v.3.01 (allocation) [28] was used as secondary data for generic materials, energy, and transport.

Table 10 shows the characteristics of the equipment used in aggregate production.

Through the LCA, a comparison of the environmental impact caused in the production of the different aggregates was carried out. The results of the characterization for the production of 1 t of aggregates are shown in Table 11. In addition, the results of variation compared to the highest value in each of the impact indicators are shown (Figure 14).

AG production resulted in a greater impact in all categories evaluated. MRA production reduced impacts such as global warming or ozone layer depletion by 41.1% and 45.9% respectively with respect to AG production. This reduction was even greater in the production of RCA, where values for these same indicators were obtained of 53.6% and 55.8%.

The increase in the environmental impact caused by the production of AG with respect to recycled aggregates could be due mainly to the greater number of processes involved in their production, to the processes of extraction and consumption of natural resources.

This analysis shows the improved environmental impact caused by the production of recycled aggregates [29,30].

To determine the environmental loads derived from the real construction of the seaport loading platform by applying MRA and RCA in the base and sub-base section (Figure 6), the CO_2_ emissions generated in the construction of the 8200 m^2^ were analyzed (Table 12). The study was carried out by applying 25 cm of MRA and 25 cm of RCA compared to a traditional construction that would correspond to 50 cm of AG applied to the entire surface of the port’s platform.

The highest value for CO_2_ emissions was for AG application in the construction of the port’s loading platform. A traditional construction would generate 19.19 tons of CO_2_. However, the application of RMA and RCA in the project section reduced CO_2_ emissions by 52.6%. This demonstrates the environmental benefit generated by using recycled aggregates in construction, which cause less impact, both in production and in its later implementation.

## 7. Conclusions

This paper provides the results of research on two recycled materials (MRA and RCA) with different composition to be used as subbases and bases of port pavements, obtaining the following conclusions:-The recycled materials analyzed, although they have somewhat lower densities than natural materials, especially the higher content of ceramic material (MRA), present physical, chemical, and mechanical properties very suitable for use as a subbase and bases for port platforms.-The process of compaction of the layers with recycled materials requires pre-wetting and a more demanding control than with the use of natural materials.-An important factor for using recycled aggregates is the content of acid-soluble sulfates. A percentage lower than 1% in SO_3_ is recommended for layers without cement treatment.-Regarding environmental issues, the leaching test classifies the RCA and MRA as inert materials.-The layers of subbase and base with recycled materials investigated do not significantly improve the equivalent modulus of the subgrade when it has good properties (subgrade type E3), instead they contribute to providing homogeneity in their behavior against the deflectometer impact test.-The recycled materials that have been used as a subbase layer (MRA) and base (RCA) have not only provided proper mechanical and deformational capabilities for use in the port pavement, but also an improvement in their properties over time.-The production of MRA and RCA applied at the base and sub-base of the loading dock generates lower impacts than the production of AG as a result of the lower number of processes required for their treatment and production.-The application of recycled materials (MRA and RCA) as base and sub-base layers, in addition to providing appropriate technical characteristics, leads to a large reduction in CO2 emissions in relation to the application of AG in these layers.

Therefore, it can be concluded that it is possible to use recycled aggregates in the construction of subbases and base layers of port pavement, providing an important environmental benefit, since it reduces the consumption of natural resources or raw materials, decreasing the ecological footprint.

## Figures and Tables

**Figure 1 materials-13-02651-f001:**
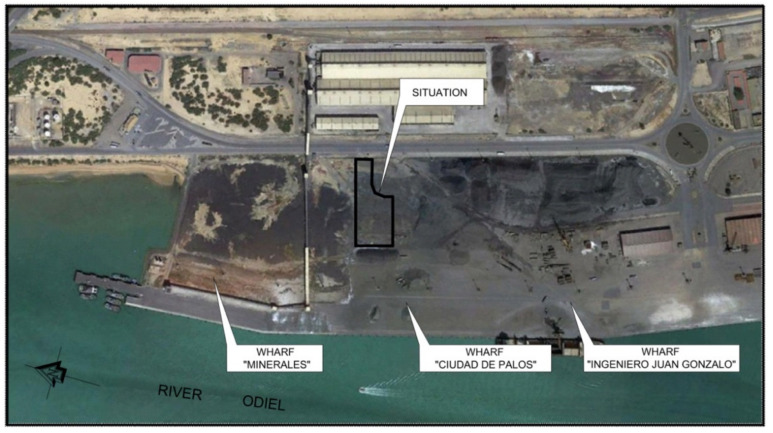
General view of the test area.

**Figure 2 materials-13-02651-f002:**
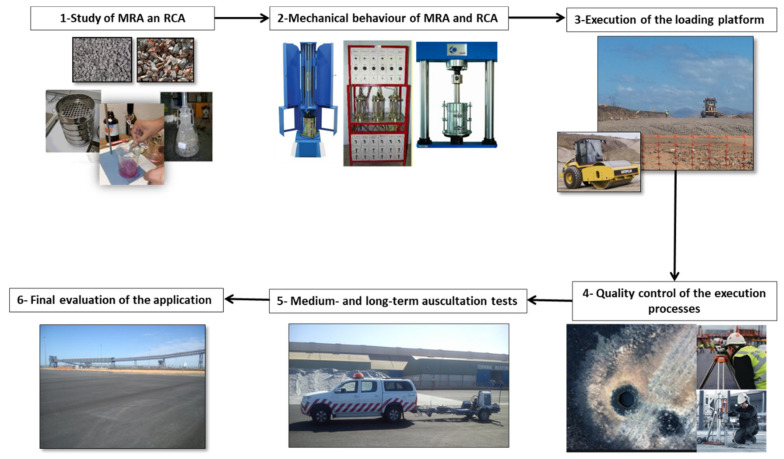
Research design.

**Figure 3 materials-13-02651-f003:**
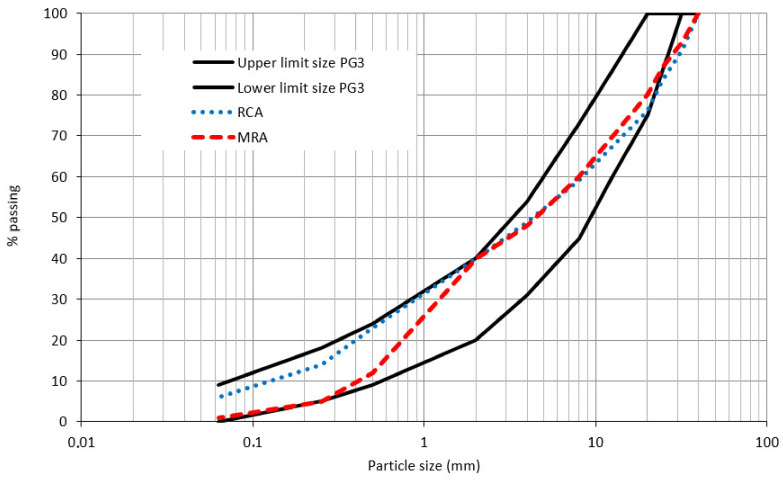
Particles’ size distribution curves compared with the particle size distribution limits.

**Figure 4 materials-13-02651-f004:**
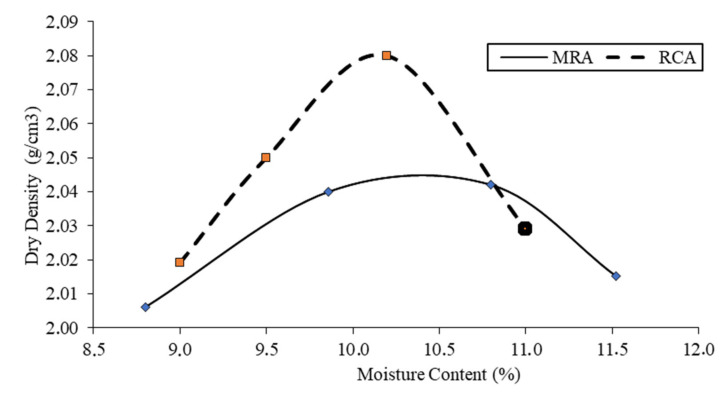
Moisture—dry density ratio.

**Figure 5 materials-13-02651-f005:**
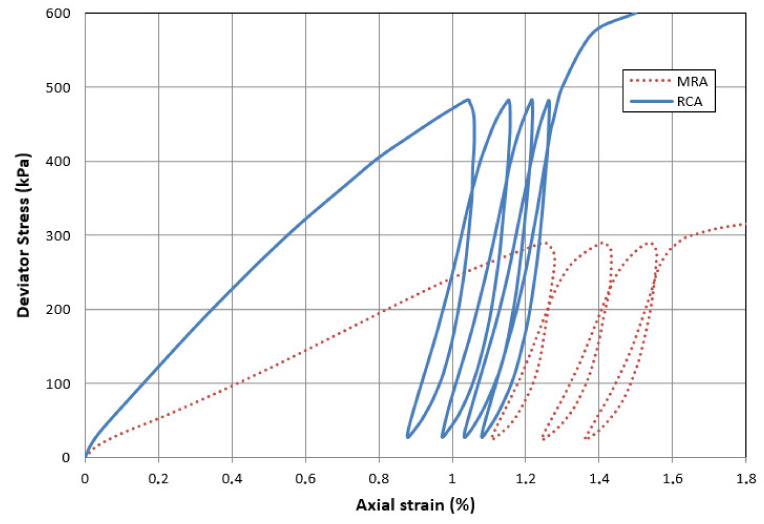
Strain–stress curves.

**Figure 6 materials-13-02651-f006:**
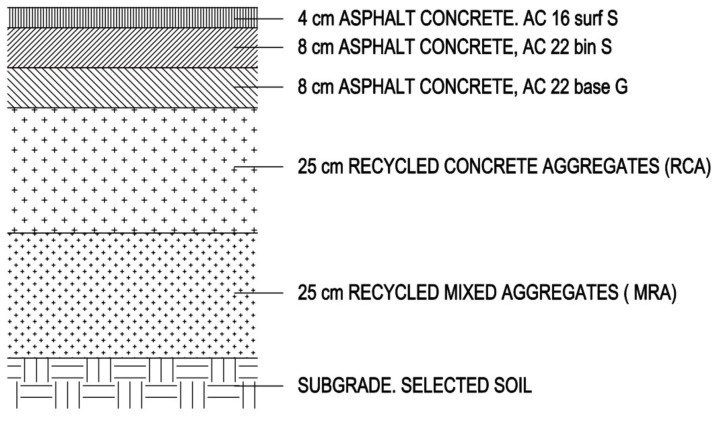
Cross sections of the port pavement.

**Figure 7 materials-13-02651-f007:**
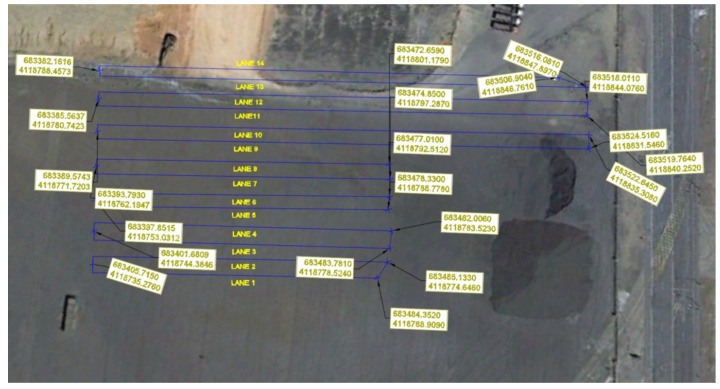
Lanes of the experimental platform.

**Figure 8 materials-13-02651-f008:**
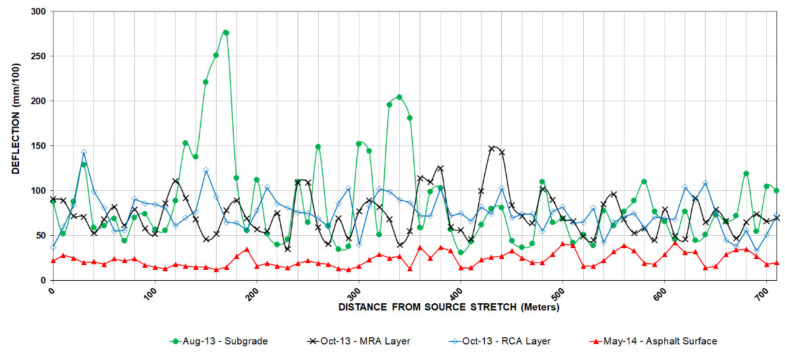
Deflection measurement.

**Figure 9 materials-13-02651-f009:**
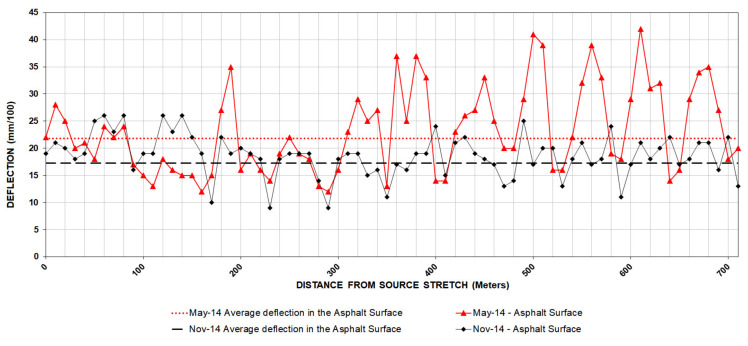
Deflection measurement in the asphalt concrete layers.

**Figure 10 materials-13-02651-f010:**
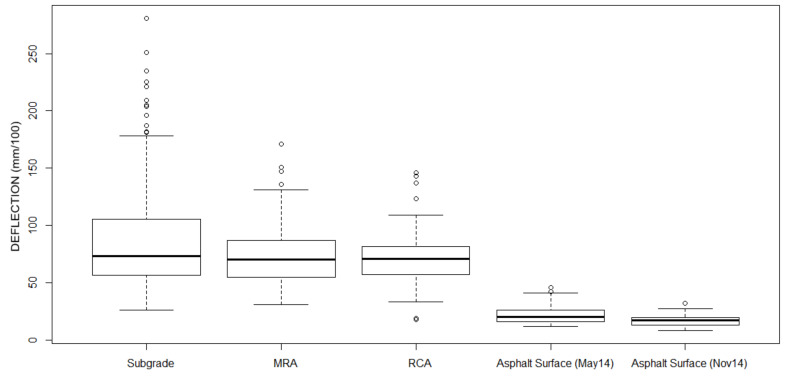
Boxplots showing deflection measurement in different layers.

**Figure 11 materials-13-02651-f011:**
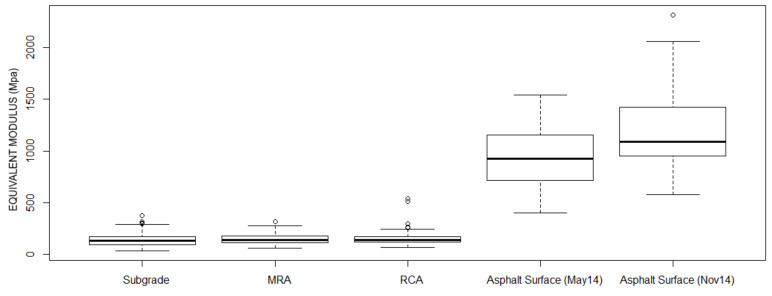
Boxplots showing evolution of the equivalent modulus in different layers.

**Figure 12 materials-13-02651-f012:**
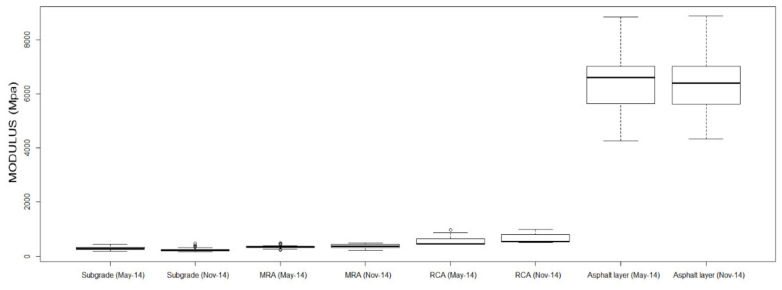
Boxplot of moduli back calculation FWD tests, 2014.

**Figure 13 materials-13-02651-f013:**
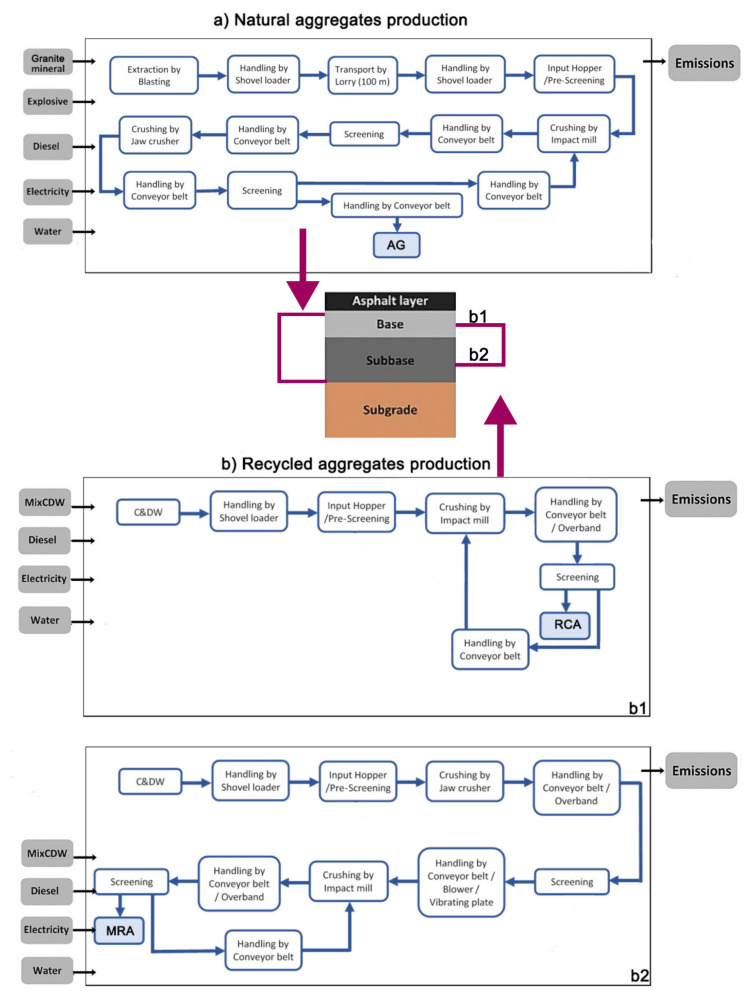
System boundaries for production of artificial gravel (AG), production of concrete and mix CDW recycled aggregate (RCA) and (MRA).

**Figure 14 materials-13-02651-f014:**
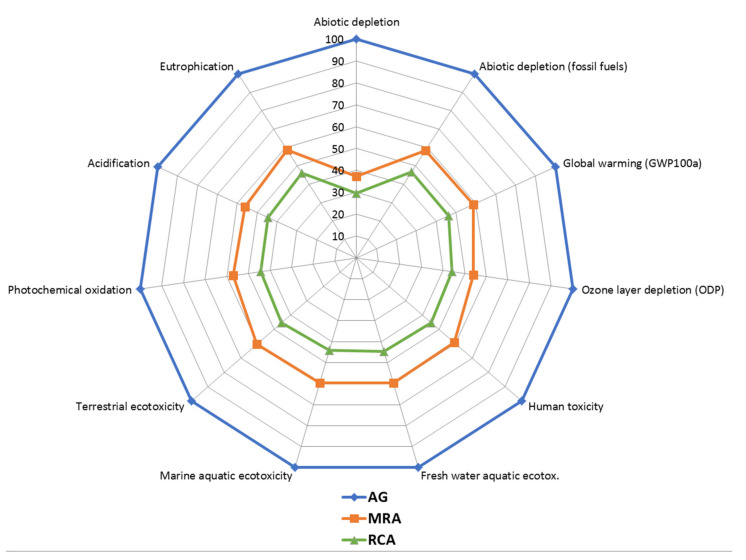
Comparative graph of aggregates production impact assessment.

**Table 1 materials-13-02651-t001:** Classification of RA proposed for international application in road sections [12].

Types of RA Proposed	Composition *				Minimum Density (SSD)	Water Absorption Capacity (%)	Los Angeles Value (%)	Water-Soluble Sulphate (%)	Proposed Uses in Road Layers
	Rc+Ru (%)	Rb (%)	Ra (%)	Others (%)					
**RCA-I**	>90	<10	<5	<1	<0.7	<6	<35	<0.7	Concrete pavement, cement treated, or unbound granular subbases
**RCA-II**	>85	<15	<10	<3	<0.8	<8	<37	<0.8	Cement-treated or unbound granular subbases
**MRA-I**	>70	<30	<5	<5	<0.8	<8	<40	<0.8	Unbound granular subbases or capping of esplanades
**MRA-II**	>60	<40	<5	<8	<1.0	<12	<45	<1.0	Capping of esplanades or subgrades

* Rc: concrete and products thereof; Ru: unbound NA; Rb: ceramic bricks and tiles, calcium silicate masonry units; Ra: bituminous materials.

**Table 2 materials-13-02651-t002:** Constituents of recycled aggregates according to EN 933-11

	Rc (%)	Ru (%)	Rb (%)	Ra (%)	Gypsum	Others (%)	Classification
MRA	25.2	46.9	25.6	4.6	0.8	0.7	MRA-I
	72.1						
RCA	21.02	67.56	8.3	2.6	0.2	0.3	RCA-II
	88.58						

**Table 3 materials-13-02651-t003:** Physical and geometric properties of recycled aggregates

Properties	RCA	MRA	Required Limits—PG3	Test Method
Water-soluble sulphate content (SO_3_%)	0.22	0.53	0.7	EN 1744-1
Acid-soluble sulphate content (SO_3_%)	0.3	0.72	0.8	EN 1744-1
Total sulphate content (SO_3_%)	0.55	0.88	2.5	EN 1744-1
Density-SSD (kg/dm^3^)				EN 1097-6
0–4 mm	2.65	2.37	>2200	
4–31.5 mm	2.42	2.30	>2200	
Water absorption (%)				EN 1097-6
0–4 mm	3.31	9.09		
4–31.5 mm	5.59	10.79		
Plasticity	Non-plastic	Non-plastic	Non-plastic	EN ISO 17892-12
Particle size distribution (mm)	Percent passing (%)			EN 933-2
40 mm	100	100		
31.5 mm	91	93		
20 mm	76	80		
8 mm	59	60		
4 mm	49	48		
2 mm	40	40		
0.5 mm	23	12		
0.25 mm	14	3		
0.063 mm	6	1		
Flakiness index	15.3	12.4	<35	EN 933-3
Los Angeles coefficient	35.2	38	<40	EN 1097-2
Crushed and broken surfaces (%)	98.6	97.7	>40	EN 933-5
Sand equivalent	42	45	–	EN 933-8

**Table 4 materials-13-02651-t004:** Results of experimental tests performed in the laboratory

Properties	Subgrade	MRA	RCA	Test Method
**Modified Proctor**				UNE-103501
Maximun dry density (kg/dm^3^)	–	2.04	2.08	
Optimum moisture content (%)	–	10.80	9.77	
**Normal Proctor**				UNE 103500
Maximun dry density (kg/dm^3^)	1.79	–	–	
Optimum moisture content (%)	5.5	–	–	
**C.B.R. index**	28	50	74	EN 13286-47

**Table 5 materials-13-02651-t005:** Results of experimental triaxial tests

Load Stage	MRA	RCA
	Secant Modulus of Deformation (MPa) (100–200 KPa)	Secant Modulus of Deformation (MPa) (150–300 KPa)
1st	24.4	48.1
2st	122.6	193.9
3st	117.8	208.3
4st	126.6	214.3

**Table 6 materials-13-02651-t006:** Leachate concentrations (mg/kg) for RCA and RMA by EN 12457-4

L/S = 10	RCA	MRA	Limit Values Directive 2003/33/EC
	(mg/kg)	(mg/kg)	Inert (mg/kg)
Cr	0.134	0.248	0.50
Ni	0.003	0.007	0.40
Cu	0.015	0.044	2.00
Zn	0.001	0.028	4.00
As	0.000	0.002	0.50
Se	0.000	0.008	0.10
Mo	0.056	0.112	0.50
Cd	0.000	0.000	0.04
Sb	0.000	0.021	0.06
Ba	0.076	0.185	20.00
Hg	0.000	0.000	0.01
Pb	0.000	0.000	0.50
Sulfates	398.0	863.0	1000

**Table 7 materials-13-02651-t007:** In situ assessments of density and moisture

Properties	Subgrade	MRA	RCA
**Density (kg/dm^3^)**			
Mean	1.79	2.08	2.12
SD	20	50	60
**Compaction (%)**			
Mean	100.0	101.7	102.1
SD	1.11	2.59	2.95
**Moisture content (%)**			
Mean	5.81	10.75	10.30
SD	1.75	0.95	0.80

**Table 8 materials-13-02651-t008:** Statistical parameters of modulus equivalent, *Ev*

Properties	Subgrade (Aug-13)	MRA (Oct-13)	RCA (Oct-13)	Asphalt Surface (May-14)	Asphalt Surface (Nov-14)
**Equivalent modulus**					
Mean (MPa)	140.3	145.6	152.2	927.1	1178.5
RSD (%)	45.5	33.6	41.7	30.0	28.5

**Table 9 materials-13-02651-t009:** Theoretical mechanical property and statistical parameter results

Layer	Date	Mean (MPa)	RSD (%)	Eck (MPa)	Thickness (m)	Theoretical Elastic Modulus (MPa)	Theoretical Deflection m/10^−6^
Asphalt Layer	May-14	6480	16.5	5409	0.20	5600	410
	Nov-14	6364	16.0	5346			
Granular Base (RCA)	May-14	556	23.6	424	0.25	350	680
	Nov-14	652	20.7	516			
Subbase (MRA)	May-14	356	18.5	290	0.25	165	1110
	Nov-14	380	18.2	311			
Subgrade	May-14	305	19.3	246	2	55	
	Nov-14	240	19.6	192			

**Table 10 materials-13-02651-t010:** Specifications of AG, RCA, and MRA production equipment

Process		Equipment	Amount	Power (kW)	Production (t/h)	Distance (km)
**AG**	Handling	Shovel loader	2	–	32.318	–
	Transport	Lorry 28 t	1	–	–	0.3
	Handling	Conveyor belt, 10 m	3	8	108.53	–
		Conveyor belt, 25 m	2	20	166.91	–
	Screening	Vibrating screen	3	18.5	225	–
	Crushing	Impact mill	1	122.06	400	–
		Jaw crusher	1	203.12	400	–
**RCA**	Handling	Shovel loader	1	–	32.318	–
	Crushing	Impact mill	1	75	250	–
	Handling	Conveyor belt, 5 m	2	4	112.1	–
		Overband	1	3.68	114.4	–
	Screening	Vibrating screen	1	22.08	250	–
**MRA**	Handling	Shovel loader	1	–	32.318	–
	Screening	Vibrating screen	3	22.08	250	–
		Vibrating plate	1	3	80	–
	Crushing	Jaw crusher	1	160	325	–
		Impact mill	1	75	250	–
	Handling	Conveyor belt, 5 m	3	4	112.2	–
		Conveyor belt, 10 m	1	7.36	112.2	–
		Overband	2	3.68	114.4	–
		Blower	1	14	144.74	–

**Table 11 materials-13-02651-t011:** Characterization results of AG, RCA, and MRA (per 1 t)

Impact Category	Units	AG	MRA	(∆%)	RCA	(∆%)
Abiotic depletion	kg Sb eq.	2.32 × 10^−6^	8.63 × 10^−7^	−62.9	6.85 × 10^−7^	−70.5
Abiotic depletion (fossil fuels)	MJ	23.9	13.9	−41.6	11.1	−53.3
Global warming (GWP100a)	kg CO_2_ eq.	2.02	1.19	−41.1	0.938	−53.6
Ozone layer depletion (ODP)	kg CFC-11 eq.	1.99 × 10^−7^	1.08 × 10^−7^	−45.9	8.79 × 10^−8^	−55.8
Human toxicity	kg 1,4-DB eq.	0.761	0.451	−40.7	0.345	−54.7
Fresh water aquatic ecotoxicity	kg 1,4-DB eq.	0.677	0.404	−40.2	0.302	−55.3
Marine aquatic ecotoxicity	kg 1,4-DB eq.	2.58 × 10^3^	1.54 × 10^3^	−40.3	1.14 × 10^3^	−55.8
Terrestrial ecotoxicity	kg 1,4-DB eq.	5.13 × 10^−3^	3.09 × 10^−3^	−39.6	2.32 × 10^−3^	−54.7
Photochemical oxidation	kg C_2_H_4_ eq.	4.43 × 10^−4^	2.53 × 10^−4^	−42.9	1.96 × 10^−4^	−55.7
Acidification	kg SO_2_ eq.	1.24 × 10^−2^	6.95 × 10^−3^	−43.8	5.49 × 10^−3^	−55.6
Eutrophication	kg PO_4_ eq.	3.82 × 10^−3^	2.24 × 10^−3^	−41.4	1.76 × 10^−3^	−53.9

**Table 12 materials-13-02651-t012:** CO_2_ emissions generated by construction of 8200 m^2^ port loading platform with RCA and MRA compared to traditional construction using AG

Impact Category	Units	Traditional Construction 50 cm (AG)	25 cm of Mixed Recycled Aggregate (MRA) and 25 cm of Recycled Concrete Aggregate (RCA)	(∆%)
Global warming (GWP100a)	kg CO_2_ eq.	1.92 × 10^4^	9.11 × 10^3^	−52.6%

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
