# Peer review of "Complete Real-Scale Application of Recycled Aggregates in a Port Loading Platform in Huelva, Spain"

_materials, 2020, doi:10.3390/ma13112651_

Round 1

Reviewer 1 Report

The article „Complete real-scale application of recycled aggregates in a port loading platform in Huelva, Spain” describes mechanical properties of both Recycled Concrete Aggregates (RCA) and Mixed Recycled Aggregates (MRA) – containing ceramic, asphalt and concrete particles. I find this article especially valuable because of description of both laboratory and lagre scale tests. Especially the latter is rarely seen in research papers. What is more the article inscribes into the idea of sustainable developement and into circular economy idea. The topic is not new, but it is very good approached. In my opinion the experimental plan is nicely designed and conducted. The conclusions are clearly stated. The only remarki is that the Introduction section is a bit to short. It should be improved and extended. I suggest to accept the paper after extending of the Introduction section.

Author Response

The article "Complete real-scale application of recycled aggregates in a port loading platform in Huelva, Spain” describes mechanical properties of both Recycled Concrete Aggregates (RCA) and Mixed Recycled Aggregates (MRA) - containing ceramic, asphalt and concrete particles. I find this article especially valuable because of description of both laboratory and large scale tests. Especially the latter is rarely seen in research papers. What is more the article inscribes into the idea of ​​sustainable development and into circular economy idea. The topic is not new, but it is very good approached. In my opinion the experimental plan is nicely designed and conducted. The conclusions are clearly stated. The only remark is that the Introduction section is a bit to short. It should be improved and extended. I suggest to accept the paper after extending the Introduction section.

Thank you very much for your comments. The introduction has been improved and extended.

For example:

“An important value is that obtained with the Los Angeles Abrasion Test, which consists of producing an abrasive action by using standard steel balls that, when mixed with aggregates and rotate in a drum during a specific number of revolutions, also cause impact on aggregates. The percentage of wear of the aggregates due to friction with steel balls is determined and is known as the Los Angeles abrasion value, it is the difference in size before and after abrasion. The lower the percentage, the better the abrasion performance of the material”.

Reviewer 2 Report

The article presents an interesting applied case study on replacing commonly used artificaial gravel by eco-friendly recycled aggregates in pavement construction. However, the text contain a number of small errors that have to be fixed, please check the following:

  • Please explain the following sentence “Recycled Aggregates which contain concrete particles larger than 90% or 85%.” I expected a size unit not %. Line 39
  • What is “R” in table 1?
  • Please explain the Los Angeles value from table 1
  • What are PG3 requirements? Please explain or add a reference.
  • Table 1 is not well edited
  • Place the 3 into superscript position in “ton/m3”, line 193 and 194
  • CBR index is shown in Table 4 not 3
  • Text in figure 12 is too small
  • Text in Figure 13 is invisible
  • Please explain the artificial gravel (AG) abbreviation as soon as it appears in the text, line 383
  • Since there is no data for Diesel Consumption in Table 10 it can be removed
  • Please move 3 to subscript position in “SO3”, line 455
  • Please check the font in the references, sometimes there are non-letter symbols instead of letters
  • Please edit the reference 16 and 20
  • Please remove the line 564
  • Please apply correct format of the references

Author Response

The article presents an interesting applied case study on replacing commonly used artificial gravel by eco-friendly recycled aggregates in pavement construction. However, the text contains a number of small errors that have to be fixed, please check the following:

  • Poitn 1: Please explain the following sentence “Recycled Aggregates which contain concrete particles larger than 90% or 85%.” I expected a size unit not %.

We agree with your comments. It has been explained: 85% or 90% corresponds to the percentage of concrete particles over 100% of the material.

  • Point 2: What is “R” in table 1?

In accordance with the UNE-EN 933-11: 2009 / AC: 2010 standard "Tests for geometrical properties of aggregates - Part 11: Classification test for the constituents of coarse recycled aggregate" it is used:

Rc: concrete and products thereof.

Ru: unbound natural aggregates.

Rb: ceramic bricks and tiles, calcium silicate masonry units.

Ra: bituminous materials.

It is explained at the end of the table

  • Point 3: Please explain the Los Angeles value from table 1

Thank you very much for your comment, the value of abrasion has been explained in the text. (Line 63-69)

“An important value is that obtained with the Los Angeles Abrasion Test, which consists of producing an abrasive action by using standard steel balls that, when mixed with aggregates and rotate in a drum during a specific number of revolutions, also cause impact on aggregates. The percentage of wear of the aggregates due to friction with steel balls is determined and is known as the Los Angeles abrasion value, it is the difference in size before and after abrasion. The lower the percentage, the better the abrasion performance of the material”.

  • Point 4: What are PG3 requirements? Please explain or add a reference.

       Thanks for your comment, the PG3 requirements are included in Table 3.

  • Point 5: Table 1 is not well edited

It has been corrected

  • Point 6: Place the 3 into superscript position in “ton/m3”, line 193 and 194

It has been corrected (ton/m3)

  • Point 7: CBR index is shown in Table 4 not 3

It has been corrected

  • Point 8: Text in figure 12 is too small and text in Figure 13 is invisible

We agree with your comments. The figures have been enlarged for a correct reading and also, it is attached in image so that the editor can put it in the size and quality that he needs.

  • Point 9 Please explain the artificial gravel (AG) abbreviation as soon as it appears in the text, line 383.

Thank you very much for your comment, the abbreviation for artificial gravel has been explained in the text.

  • Point 10: Since there is no data for Diesel Consumption in Table 10 it can be removed

We agree with your comments. The diesel column has been removed from table 10

  • Point 11: Please move 3 to subscript position in “SO3”, line 455

It has been corrected

  • Point 12: Please edit the reference 16 and 20

Thank you very much for your comment. both references have been corrected and edited.

  • Point 13: Please remove the line 564

The line has been removed

Reviewer 3 Report

This study investigated the use of recycled aggregates for road based and sub-bases construction. Comprehensive laboratory experiments and field tests were conducted to assess the properties and sustainability of the RCA and MRA for their application on the base and subbases construction, respectively.  The results showed that the recycled shows good performance and can potentially be used as the construction materials for the subbases and base layers of the pavement to reduce the consumption of natural resources and CO2 emission. The manuscript can be considered publish after some minor revisions:

  1. Line 205 CBR results are shown in Table 4 not Table 3
  2. Line 231 secant deformation modulus are shown in Table 5
  3. Line 298 to 300, please demonstrated the reasons for less deflection in November than that in May.

Author Response

This study investigated the use of recycled aggregates for road based and sub-bases construction. Comprehensive laboratory experiments and field tests were conducted to assess the properties and sustainability of the RCA and MRA for their application on the base and subbases construction, respectively.  The results showed that the recycled shows good performance and can potentially be used as the construction materials for the subbases and base layers of the pavement to reduce the consumption of natural resources and CO2 emission. The manuscript can be considered publish after some minor revisions:

  • Point 1: Line 205 CBR results are shown in Table 4 not Table 3

Thank you very much for your comment. It has been corrected

  • Point 2: Line 231 secant deformation modulus are shown in Table 5

We agree with your comments. It has been corrected

  • Point 3: Line 298 to 300, please demonstrated the reasons for less deflection in November than that in May.

According to other studies, RCA has cementitious properties, mainly due to the non-hydrated cement particles. For this reason, the properties improve over time.

Reviewer 4 Report

The authors of the article sought the answer to the question whether two types of recycled aggregates (Recycled Concrete Aggregates (RCA) and Mixed Recycled Aggregates (MRA) - a mix of ceramic, asphalt and concrete particles) can be used as a replacement for traditional aggregates in structural layers of platforms seaports.

The topic is important and interesting, because this approach has an ecological aspect, allows you to use waste that causes environmental problems. At the same time, it can reduce the consumption of natural aggregates, which (according to forecasts) may soon be lacking in some countries. In the face of the current global situation - transporting aggregates from other countries can also be difficult and associated with high costs.

Possibilities of using recycled aggregates have already been undertaken by other researchers, but in the case of the reviewed article originality is related to the application carried out at a specific location and confirmation of the possibility of using aggregates on a large scale. The article may be a source of knowledge for other researchers in this field. The article was written in a legible way, enabling understanding of research topics.

The authors first conducted laboratory tests of recycled aggregates. After obtaining positive results, they put them into practice. Tests conducted during the construction of the platform and after its completion were carefully planned, the photo documentation and description was included in the article.

The conclusions are in accordance with the research results obtained by the authors and confirm the possibility of using recycled aggregates as a replacement for natural aggregates in the structural layers of seaport platforms.

Author Response

The authors of the article sought the answer to the question whether two types of recycled aggregates (Recycled Concrete Aggregates (RCA) and Mixed Recycled Aggregates (MRA) - a mix of ceramic, asphalt and concrete particles) can be used as a replacement for traditional aggregates in structural layers of platforms seaports.

The topic is important and interesting, because this approach has an ecological aspect, allows you to use waste that causes environmental problems. At the same time, it can reduce the consumption of natural aggregates, which (according to forecasts) may soon be lacking in some countries. In the face of the current global situation - transporting aggregates from other countries can also be difficult and associated with high costs.

Possibilities of using recycled aggregates have already been undertaken by other researchers, but in the case of the reviewed article originality is related to the application carried out at a specific location and confirmation of the possibility of using aggregates on a large scale. The article may be a source of knowledge for other researchers in this field. The article was written in a legible way, enabling understanding of research topics.

The authors first conducted laboratory tests of recycled aggregates. After obtaining positive results, they put them into practice. Tests conducted during the construction of the platform and after its completion were carefully planned, the photo documentation and description was included in the article.

The conclusions are in accordance with the research results obtained by the authors and confirm the possibility of using recycled aggregates as a replacement for natural aggregates in the structural layers of seaport platforms.

Thank you very much for your comment